# FrugalGPT: How to Use Large Language Models While Reducing Cost and Improving Performance

## Abstract

The rapid adoption of large language models (LLMs) has led to an growing number of companies offering generative LLMs as callable services at varying costs. We find that popular generative LLM APIs, such as GPT-4, ChatGPT, and J1-Jumbo, exhibit heterogeneous pricing structures, with fees that can differ by two orders of magnitude, and heterogeneous performance across tasks and input queries. This makes it challenging for users to decide which generative LLM APIs to utilize for their applications and budget. Motivated by these findings, we propose FrugalGPT, an algorithmic framework that adaptively selects which generative LLMs to use for different queries to reduce cost and improve accuracy. Our experiments demonstrate that, for a range of natural language tasks including news classification, reading comprehension, and scientific question answering, FrugalGPT can match the performance of the best individual generative LLM (e.g., GPT-4) with up to a 98% cost reduction or improve the accuracy over GPT-4 by 4% at the same cost. The ideas and findings presented in this paper lay a foundation for using LLMs sustainably and efficiently.

## 1 Introduction

We are currently witnessing a surge in the adoption of large language models (LLMs). The enticing potential of employing generative LLMs for applications in commerce, science, and finance has led to a growing number of companies (such as OpenAI, AI21, CoHere, etc.) offering generative LLMs as callable services. Consequently, machine learning (ML) practitioners now frequently build applications by invoking these foundation models.

However, users often face challenges in deciding which generative LLM APIs to utilize for their applications and budget. The cost of generative LLMs can vary by up to two orders of magnitude: for instance, the prompt cost for 10M tokens is $30 for OpenAI's GPT-4 but only $0.2 for GPT-J hosted by Textsyth (as shown in Table 1). Smaller generative LLMs are generally more affordable, but their performance is comparatively limited (as depicted in Figure 1(d)). Larger generative LLMs like GPT-4 offer better performance but at the risk of escalating costs. In addition to the financial burden, employing larger LLMs incurs significant environmental and energy impacts Bender et al. (2021); Wu et al. (2022), affecting the social welfare of current and future generations.

In this paper, we demonstrate that it is possible to *simultaneously lower the cost and enhance the performance of generative LLM applications*. This is based on two key findings. First, no generative LLM is "universally" superior to others. Take the task of predicting price trends from news headlines as an example. There are 6% of queries where GPT-J is entirely accurate while GPT-4 makes errors, and 80% of queries where both models provide identical responses (as illustrated in Figure 1(c)). Directing 86% of queries to GPT-J and the remaining 14% to GPT-4 is considerably more cost-effective and performant than relying solely on GPT-4. Second, assessing the quality of an answer to a query is often easier than generating the answer itself. In fact, we discovered that for many natural language tasks, a "small" language model (such as DistillBERT) can accurately predict the answer quality of a large model (e.g., GPT-4).

Inspired by these findings, we propose FrugalGPT, an algorithmic framework that adaptively determines which generative LLMs to use given a user's budget. FrugalGPT first learns a generation

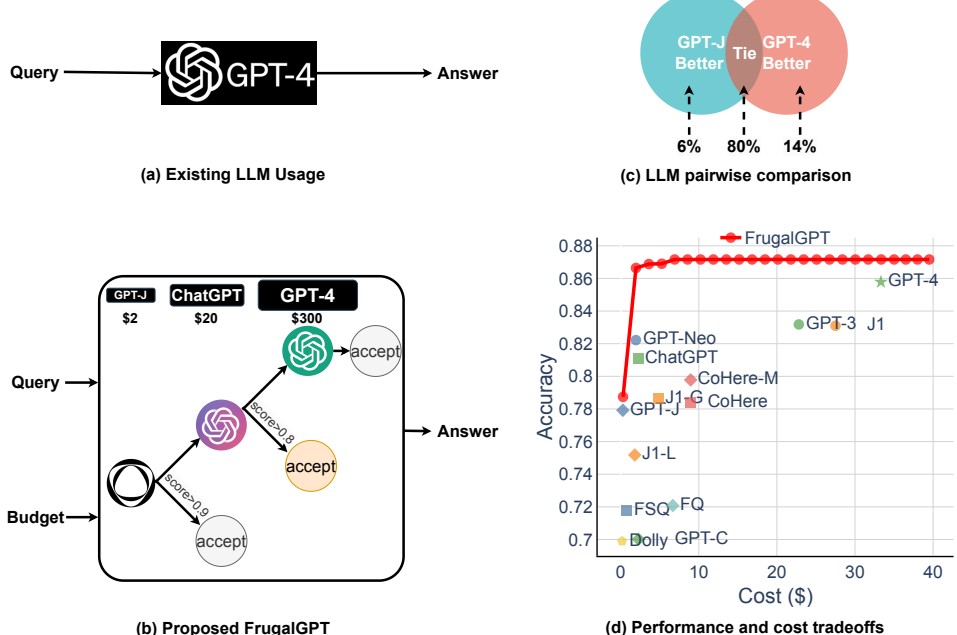

Figure 1: Comparisons of different approaches to using LLM services. (a) The standard usage sends queries to a single LLM (e.g., GPT-4), which can be expensive. (b) FrugalGPT, adaptively decides which LLMs to trigger for different user queries to reduce the inference cost. By optimizing the selection of different LLM APIs (e.g., GPT-J, ChatGPT, and GPT-4), we can achieve substantial efficiency gains. (c) LLM performance breakdown on HEADLINES (a financial news dataset). GPT-J outperforms GPT-4 on 6% of queries and produces identical generations on 80% of queries. (d) FrugalGPT can reduce the inference cost by 98% while exceeding the performance of the best individual LLM (GPT-4) on HEADLINES. This is because FrugalGPT successfully learns data subsets on which inexpensive LLMs like GPT-J are as good as or better than GPT-4, and directs these data to the corresponding low-cost LLMs only.

judger that assigns a score to indicate the quality of different LLMs' generations for any given query. It then invokes a list of LLMs sequentially until the judger's score for an answer surpasses a threshold. For example, FrugalGPT may initially call GPT-J to obtain an answer. If the judger's score for this answer is lower than a threshold of 0.9, ChatGPT is subsequently invoked to generate a new response. The judger's score for ChatGPT's answer exceeds a threshold of 0.8, so no further generative LLMs are needed, and ChatGPT's answer is returned to the user. We developed an efficient optimization technique to determine the optimal order of generative LLMs to call and the stopping threshold for each generative LLM as the core of FrugalGPT.

To demonstrate the potential of FrugalGPT, we implement and evaluate it on various tasks, such as news classification, reading comprehension, and scientific question answering, using real-world generative LLMs, including ChatGPT Cha, GPT-3 Brown et al. (2020), and GPT-4 OpenAI (2023). Our experiments show that FrugalGPT can save up to 98% of the inference cost of the best individual LLM API while matching its performance on the downstream task. On the other hand, FrugalGPT can improve performance by up to 4% at the same cost. This is because FrugalGPT accurately identifies queries on which some inexpensive LLMs are correct but the most powerful LLM (e.g., GPT-4) is incorrect, and directs these queries exclusively to the low-cost LLMs. We will also release the code and datasets used in our experiments. We hope FrugalGPT paves the way for enhancing LLMs' inference cost and performance.

**Related Works.** **Model Ensembles.** Model ensembles Dong et al. (2020), which involve combining multiple ML models for prediction, have gained popularity in supervised learning García-Pedrajas (2009); Friedman (2002), unsupervised learning Yang et al. (2014), semi-supervised learning Gupta et al. (2022), and weakly supervised learning Diba et al. (2017). Recent work Arora et al. (2022) shows that fusing multiple generations from GPT-J Wang & Komatsuzaki (2021) can compete with GPT-3's performance, and synthesizing multiple open-source LLMs' generations leads to better performance than individual LLMs Jiang et al. (2023). Model ensembles typically require

white-box access to multiple models for training, but LLM APIs are often black-box. Moreover, model ensembles necessitate querying all models for any single query, thereby increasing costs.

**ML-as-a-Service and Cascade.** Generative LLM APIs constitute a crucial component of the rapidly expanding machine-learning-as-a-service (MLaaS) industry. Recent studies have demonstrated the diversity of different ML APIs' predictions Buolamwini & Gebru (2018); Koenecke et al. (2020); Chen et al. (2021). The concept of using multiple services for speed is known as model cascade Viola & Jones (2004), which has been applied in predictive ML domains such as pedestrian detection Cai et al. (2015) and facial recognition Li et al. (2015); Sun et al. (2013). Recent work Chen et al. (2020; 2022) builds a customized cascade for cost reduction, with a focus on classification ML APIs. However, their approach needs to estimate the performance of an ML API without querying it, based on simple signals such as labels from a proxy model. Such pre-query estimation is challenging for generative LLM APIs, whose outputs encompass a much larger space. FrugalGPT overcomes this by creating a post-query quality estimator. Furthermore, for a given query, previous work invokes at most two APIs, while FrugalGPT allows invoking three or more given the vast number of LLM APIs. This renders it computationally more challenging to find the best calling strategies, and thus we also develop novel techniques to identify the optimal strategies efficiently (Section 3).

**Speculative Decoding.** Speculative decoding has recently emerged for LLM inference acceleration without retraining or model architecture modification Leviathan et al. (2023); Chen et al. (2023); Sun et al. (2023). Its goal is to provide the same output as a large LLM at lower latency. It relies on inexpensive LLMs for most generation and switches to costly LLMs when necessary. However, it requires access to the decoding module, which is not available for proprietary LLMs like GPT-4, and because it aims to give the same answer as the large LLM, it misses the opportunity to provide a better answer in cases where the small LLM is more accurate.

The remainder of the paper is organized as follows. We start by offering more context and the problem statement in Section 2. We present how FrugalGPT works in Section 3. Section 4 shows the empirical benefits of FrugalGPT using real-world LLM APIs (including GPT-3, ChatGPT, and GPT-4). We discuss future prospects in Section 5.

## 2 SCOPE AND PROBLEM STATEMENT

**Natural language query answering.** In this paper, we concentrate on the standard natural language query answering task, where the objective is to answer a query $q$ sampled from a natural language query distribution $\mathcal{Q}$. Various real-world natural language tasks, such as news classification and commonsense reasoning, can be formulated as query-answering problems.

**LLM marketplace.** We consider answering queries via the LLM market, which comprises $K$ different LLM APIs, denoted by $\{f_i(\cdot)\}_{i=1}^{K}$. Each $f_i(\cdot) : \mathcal{P} \mapsto \mathcal{A}$ is a function that, given a prompt $p$ from the prompt space $\mathcal{P}$, generates an answer from the answer distribution $\mathcal{A}$. Note that to use LLM APIs, one has to convert each query $q$ to some corresponding prompt first. LLM APIs are associated with their own *cost*, typically consisting of three components: a portion proportional to the length of the prompt, a portion proportional to the length of the generated answer, and (sometimes) a fixed cost per query. Formally, given a prompt $p$, the cost of using the $i$th LLM API is denoted by $c_i(p) \triangleq \tilde{c}_{i,2}\|f_i(p)\| + \tilde{c}_{i,1}\|p\| + \tilde{c}_{i,0}$, where $\tilde{c}_{i,j}, j = 0, 1, 2$ are constants.

**An illustrative example.** Adapting the case study provided by Kaiser & Slowik (2023), assume a small business operates a customer service using GPT-4. The company caters to 15,000 customers each month, with each customer asking three questions twice a week, totaling 360,000 queries per month. Suppose for each question, its prompt averages 1800 tokens and the answer is around 80 tokens (as estimated by Kaiser & Slowik (2023)). Considering that the input and response costs of GPT-4 are \$0.03 and \$0.06 per thousand tokens, the total monthly cost amounts to $360 \times (\$0.03 \times 1800 + \$0.06 \times 80) \approx \$21.2K$. Such a high cost is prohibitive for many small businesses.

**Problem statement: budget-aware LLM API usage.** Our primary goal in this paper is *leveraging LLM APIs within a budget constraint*. Formally, this can be formulated as maximizing the overall task performance $\mathbb{E}_{(q,a)\in\mathcal{Q}\times\mathcal{A}}[r(a, \hat{a}(s, q))]$, while ensuring the average cost is bounded by a user-defined value $b$, i.e., $\mathbb{E}_{(q,a)\in\mathcal{Q}\times\mathcal{A}}[c(s, q)] \le b$. Here, $a$ denotes the correct answer to the query

$q$, $\hat{a}(s, q)$ is the generated answer by some strategy $s$ for query $q$, and $c(s, q)$ is the associated cost for processing query $q$ using strategy $s$. The reward function $r(\cdot, \cdot)$ measures how closely the generated answer aligns with the correct one.

## 3 FRUGALGPT: A COST-AWARE PARADIGM TO LEVERAGE LLMS

In this section, we present FrugalGPT, a cost-aware approach designed to harness the power of multiple LLM services. We begin by outlining the FrugalGPT pipeline and explaining the functionality of each component. Subsequently, we delve into the construction of the FrugalGPT pipeline for a given application and user budget.

**FrugalGPT Pipeline.** FrugalGPT comprises three main components: an LLM router, an answer scorer, and a stop judger. Given a user query $q$, the LLM router is first invoked to select an LLM to obtain its response to the query. Next, the generation scorer takes the query, the answer, and the selected LLM as input and generates a quality measurement as output. Based on the quality measurement and the invoked LLM service, the stop judger determines whether (i) to stop and return the answer, or (ii) to repeat the process of invoking the LLM router and generation scorer.

The LLM router consists of two parts. First, given the previously invoked LLM service $k'$, it selects the next LLM service to use, denoted by $k \triangleq \sigma(k')$, where $\sigma : \{\varnothing, 1, 2, \cdots, K\} \mapsto \{\varnothing, 1, 2, \cdots, K\}$ is a permutation of all LLM services (with $\varnothing$ representing no invocation). Second, it sends the query $q$ to the $k$th service and obtains the generation $f_k(q)$ as output. Although the service permutation could depend on the input query in principle, our instantiation adopts a query-agnostic permutation $\sigma(\cdot)$ for simplicity.

The generation scorer, denoted by $g_i(\cdot, \cdot) : \mathcal{Q} \times \mathcal{A} \mapsto [0, 1]$, generates a quality measurement given a query and an answer produced by the $i$th LLM API. Generally, the generation scorer can be any function such that its higher values strongly correlate with the input generation's quality. In our instantiation, we adopt a DistilBERT Sanh et al. (2019) model tailored for regression as the generation scorer, as it is smaller, cheaper, and faster than LLM services while still providing a reliable quality measurement. Specifically, we have added a linear layer on top of the original DistilBERT that takes the last representation layer (768-dimensional) as input and produces a 2-dimensional output to encode the answer correctness. The maximum value of the last layer, normalized by softmax, is returned as the final score. We utilize the same embedding as DistilBERT, ensuring compatibility and seamless integration. For each LLM service, we have trained the model weights with (i) the query appended by the service's response as input features, and (ii) whether the response is correct as labels. We will present an ablation study on the generation scorer in Section 4.

The stop judger is responsible for deciding when to stop and return the answer to the user. As higher quality measurements indicate better generation quality, we use a threshold-based stop judger: return answer $a$ if the quality measurement $g_i(q, a)$ is higher than a threshold $\tau_i$ and go back to the router otherwise. The threshold vector $\tau$ controls the trade-offs between performance and cost: larger values often lead to better performance, while smaller values favor lower cost.

**Joint optimization of the FrugalGPT Pipeline.** Configuring the LLM router and stop judger appropriately is crucial to FrugalGPT. Technically, we need to configure (i) the LLM router's service permutation function $\sigma(\cdot)$ and (ii) the stop judger's threshold vector $\tau$. Our goal is to maximize the expected reward on the query distribution while satisfying the user budget. This problem can be formally modeled as the following optimization problem:

$$\max_{\sigma(\cdot), \tau} \mathbb{E}\left[r(a, f_z(q))\right] \textit{ // Performance}$$

$$s.t. \; \mathbb{E}\left[\sum_{z' : z' = \sigma^{(t')}(\varnothing), t' \leq t} \tilde{c}_{z',2} \|f_i(q)\| + \tilde{c}_{z',1}\|q\| + \tilde{c}_{z',0}\right] \leq b, \textit{ // cost bounded by budget}$$

$$t \in [L], z = \sigma^{(t)}(\varnothing), g_z(q, f_z(q)) > \tau_z, \textit{ // Stop at the t-th iteration}$$

$$\forall t' < t, z' = \sigma^{(t')}(\varnothing), g_{z'}(q, f_{z'}(q)) \leq \tau_{z'} \textit{ // No stop at previous iterations}$$

Here, the objective is the expected performance (reward), the first constraint ensures the average cost is bounded by the budget, the second constraint indicates that the stop judger returns the answer at the $t$-th iteration, and the last constraint indicates that the LLM router and the generation scorer are called repeatedly for previous iterations. $L$ is a hyperparameter that controls the maximum number of LLM services to call for a query. Solving this problem is inherently challenging because the optimization space is vastly large. $\sigma(\cdot)$ is a permutation function over all possible LLM services, and exhaustive search takes $O(L^K)$ computations. Moreover, even if $\sigma(\cdot)$ is fixed, the problem is non-convex with respect to the threshold vector $\tau$. In fact, the problem is a mixed-integer optimization in nature, which is computationally expensive to solve in general.

To overcome this computational obstacle, we design a specialized optimizer for this problem. It (i) prunes the search space of $\sigma(\cdot)$ by ignoring any consecutive selection of LLMs with small answer disagreement, and (ii) approximates the objective by interpolating it within a few samples.

Search space pruning removes candidate permutation functions with relatively small maximum performance improvement, or *MPI*. Here, *MPI* is a function of two LLMs, $k_1, k_2$, that measures at most how many mistakes $k_2$ incurs can be fixed by $k_1$. Formally, $MPI(k_1, k_2) \triangleq \Pr[r(q, f_{k_1}(q)) > r(q, f_{k_2}(q))]$. Suppose $k$ is called from the last iteration in the cascade. Then in the next iteration, calling any LLMs with small MPI would not yield significant performance gains and thus could be avoided. Inspired by this, we introduce the following pruning condition

$$\sigma(k) \in \{k' \in K \mid MPI(k'', k) \geq MPI(k', k) \text{ for at most } m - 1 \text{ other values of } k'' \in K\}$$

That is to say, given the previously invoked LLM $k$, the next LLM to call must hold the top-$m$ value of MPI with respect to $k$. This reduces the search complexity from $O(L^K)$ to $O(L^m)$. In practice, we found that $m = 3$ often suffices to identify a high-quality cascade.

Now suppose the function $\sigma(\cdot)$ is fixed. The remaining step is to find the optimal threshold vector $\tau$. This can be resolved via a two-stage approximation. First, we divide the search space $[0, 1]^L$ into a few equal-size grids. Next, within each grid, we approximate the objective by a quadratic function of the threshold vector, whose parameters are determined by the grid bound values. Then within each grid, we can leverage a QP solver to find the optimal solution. The final solution is the best among all grids. The combination of the above two techniques provides an efficient implementation with satisfactory performance, as demonstrated later in Figure 3.

## 4  EXPERIMENTS

In this section, we present an empirical study on FrugalGPT. Our goals are four-fold: (i) understand when and why FrugalGPT lowers the cost, (ii) quantify the cost savings attained by FrugalGPT while matching the best individual LLM API's performance, (iii) measure the trade-offs between performance and cost enabled by FrugalGPT, and (iv) explore how different factors including data distribution shifts and scorer's quality affect FrugalGPT.

**Setups: LLM APIs, Tasks, Datasets, and FrugalGPT instances.**  We have selected 14 LLM APIs from 6 mainstream providers, namely, OpenAI Ope, AI21 AI2, CoHere CoH, Textsynth Tex, Databricks Dol, and ForeFrontAI FFA. The details are summarized in Table 1. FrugalGPT has been developed on top of these APIs and evaluated on a range of datasets belonging to different tasks, including HEADLINES Sinha & Khandait (2021), OVERRULING Zheng et al. (2021), COQA Reddy et al. (2019), AGNEWS Zhang et al. (2015) and SCIQ Welbl et al. (2017). More details of the datasets and tasks can be found in the Appendix. We focus on FrugalGPT with the hyperparameter $L = 3$, as this simplifies the optimization space and demonstrates exciting results. Each dataset is randomly split into a training set (50%) to learn FrugalGPT and a test set for evaluation (50%).

**A Case Study.**  We begin with a case study on the HEADLINES dataset. We set the budget to be $6.5, which is one-fifth of GPT-4's cost. As depicted in Figure 2 (a), the learned FrugalGPT sequentially calls GPT-J, J1-L, and GPT-4. For any given query, it first extracts an answer from GPT-J. If the score of this answer is greater than 0.96, the answer is accepted as the final response. Otherwise, J1-L is queried. J1-L's answer is accepted as the final response if its score is greater than 0.37; otherwise, GPT-4 is invoked to obtain the final answer. Interestingly, this approach outperforms GPT-4 for numerous queries. For instance, given a headline "Gold prices trade near 3-month high as Fed

Table 1: Summary of commercial LLM APIs. We use 14 LLM APIs from 6 providers. The cost was retrieved in March 2023. The cost can have three additive components: input (proportional to the number of input tokens), output (proportional to the number of generated tokens) and a fixed cost per request. The LLMs's costs can differ by up to 2 orders of magnitudes. For example, to process 10M input tokens, GPT-J from Textsynth costs only $0.2, but OpenAI's GPT-4 needs $30.

| Provider | API | Size/B | Cost (USD) | | |
|---|---|---|---|---|---|
| | | | 10M input tok. | 10M output tok. | request |
| OpenAI | GPT-Curie | 6.7 | 2 | 2 | 0 |
| | ChatGPT | NA | 2 | 2 | 0 |
| | GPT-3 | 175 | 20 | 20 | 0 |
| | GPT-4 | NA | 30 | 60 | 0 |
| AI21 | J1-Large | 7.5 | 0 | 30 | 0.0003 |
| | J1-Grande | 17 | 0 | 80 | 0.0008 |
| | J1-Jumbo | 178 | 0 | 250 | 0.005 |
| Cohere | Xlarge | 52 | 10 | 10 | 0 |
| | Medium | 6.1 | 10 | 10 | 0 |
| Textsynth | GPT-J | 6 | 0.2 | 5 | 0 |
| | FAIRSEQ | 13 | 0.6 | 15 | 0 |
| | GPT-Neox | 20 | 1.4 | 35 | 0 |
| Databricks Model Serving | Dolly | 7 | 0.27 | 0.27 | 0 |
| ForeFrontAI | QA | 16 | 5.8 | 5.8 | 0 |

Table 2: Cost (USD) savings by FrugalGPT to match the best individual LLM's performance.

| Dataset | Best individual LLM | Cost to reach the same accuracy | | Cost Savings |
|---|---|---|---|---|
| | | Best individual LLM | FrugalGPT | |
| HEADLINES | GPT-4 | 33.1 | 0.6 | 98.3% |
| OVERRULING | GPT-4 | 9.7 | 2.6 | 73.3% |
| COQA | GPT-3 | 72.5 | 29.6 | 59.2% |
| AGNEWS | GPT-4 | 64.6 | 15.9 | 75.4% |
| SCIQ | GPT-3 | 132.4 | 63.1 | 52.3% |

begins meeting" from NASDAQ, FrugalGPT accurately predicts that the price is going down, while GPT-4 provides an incorrect answer (as shown in Figure 2(b)). Overall, FrugalGPT results in both accuracy gains and cost reduction, as illustrated in Figure 2(c).

**LLM diversity.** Why can multiple LLM APIs potentially produce better performance than the best individual LLM? This is often due to generation diversity: even an inexpensive LLM can sometimes correctly answer queries on which a more expensive LLM fails. Recall that we introduce maximum performance improvement (*MPI*) in Section 3 as an pruning metric. In fact, it also measures the generation diversity well: larger value of MPI indicates that one generative LLM give more responses different from another one. As shown in Figure 2 (d), MPI is indeed large for many pairs of generative LLMs. For instance, there are 6% queries where GPT-4 is incorrect but GPT-J (and GPT-C, J1-L, or Dolly) can give desired answers. This indicates the potential of combining multiple generative LLMs, and verifies why FrugalGPT offers cost reduction without performance drops.

**Cost Savings.** Subsequently, we examine if FrugalGPT can reduce costs while maintaining accuracy and, if so, by how much. Table 2 displays the overall cost savings of FrugalGPT, which range from 50% to 98%. This is feasible because FrugalGPT identifies the queries that can be accurately

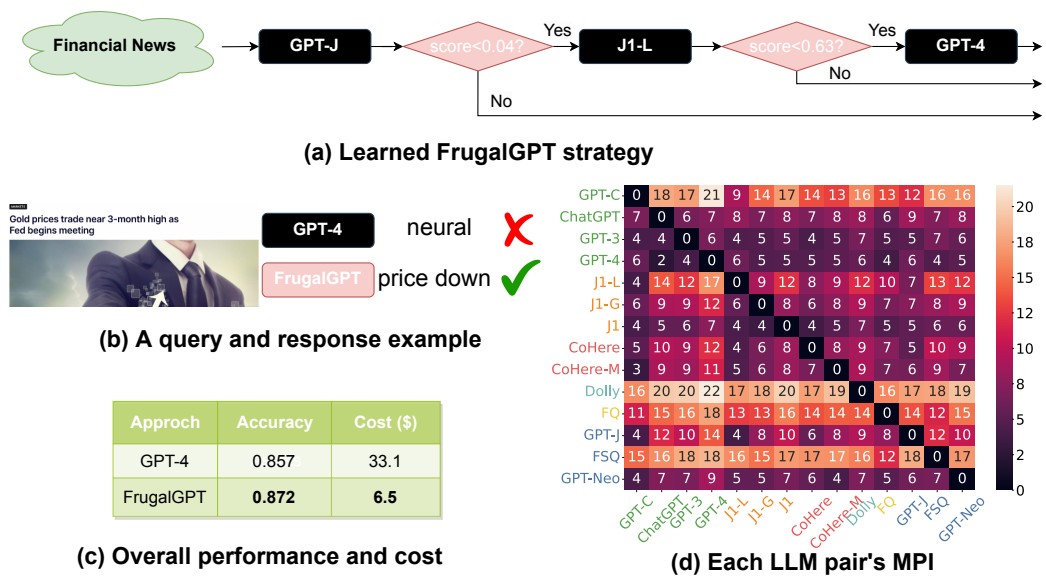

Figure 2: A case study of FrugalGPT on the HEADLINES dataset. (a) The cascade strategy that FrugalGPT learned on this dataset with an overall budget of $6.5, one-fifth of GPT-4's cost. FrugalGPT avoids querying GPT-4 as long as GPT-J and J1-L produce high-quality answers. (b) Sometimes GPT-4 makes a mistake, but FrugalGPT learns to use the correct answers by J-1 and GPT-J. (c) Overall, FrugalGPT reduces the cost by 80%, while improving the accuracy by 1.5% compared to GPT-4. (d) The maximum possible improvement (MPI) for each LLM pair, measuring how often one LLM (each row) makes a mistake while another (each column) is correct. Even for the best individual LLM, GPT-4, cheap LLMs (e.g., GPT-J) can be better on 6% of the data.

answered by smaller LLMs and, as a result, only invokes those cost-effective LLMs. Powerful but expensive LLMs, such as GPT-4, are utilized only for challenging queries detected by FrugalGPT.

**Performance and Cost Trade-offs.** Now, we investigate the trade-offs between performance and cost achieved by FrugalGPT, as illustrated in Figure 3. Here we focus on three datasets due to space limitations; more results on other datasets can be found in the Appendix.

Several interesting observations can be made. First, the cost ranking of different LLM APIs is not fixed. For instance, J1 is the second most expensive LLM on the HEADLINES dataset, while GPT-3 holds that position on the OVERRULING and COQA datasets. This is primarily due to the heterogeneous pricing mechanism: J1 incurs a high cost for each generated token but charges nothing for input tokens, whereas GPT-3 charges for both input and output tokens. Moreover, more expensive LLM APIs sometimes result in worse performance than their cheaper counterparts. For example, J1 is costlier than GPT-3 on HEADLINES, but its performance is inferior. These observations underscore the importance of aptly selecting LLM APIs, even in the absence of budget constraints.

Next, we note that FrugalGPT enables smooth performance-cost trade-offs across all evaluated datasets. This offers flexible choices to LLM users and potentially helps LLM API providers save energy and reduce carbon emissions. In fact, FrugalGPT can simultaneously reduce costs and improve accuracy. For example, on the OVERRULING dataset, FrugalGPT achieves a 1% accuracy gain while reducing costs by 73% compared to the best LLM API, GPT-4. This is likely because FrugalGPT integrates knowledge from multiple LLMs.

The example queries shown in Figure 3 further aid in understanding why FrugalGPT can simultaneously improve performance and reduce costs. GPT-4 makes mistakes on some queries (e.g., the first example in part (a)), but some low-cost APIs provide correct predictions. FrugalGPT accurately identifies those queries and relies solely on the inexpensive APIs. For example, GPT-4 incorrectly infers no overruling from the legal statement "The time has come to reconcile and regularize our cases in this field," as shown in Figure 3(b). However, FrugalGPT accepts GPT-J's correct answer,

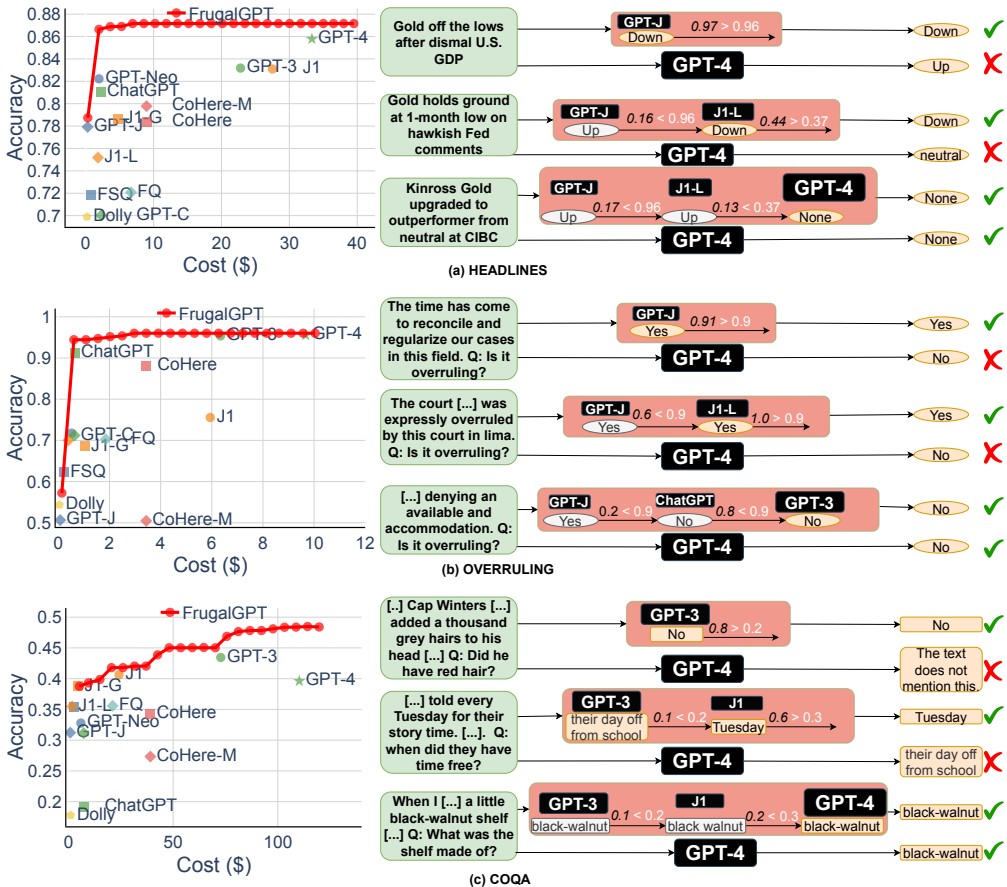

Figure 3: Accuracy and cost tradeoffs achieved by FrugalGPT. Overall, FrugalGPT often achieves the same performance of the best individual LLM API (e.g., GPT-4) with orders of magnitudes smaller cost. When incurring the same cost, FrugalGPT can improve the accuracy by up to 5%. Examples of FrugalGPT for each dataset are shown on the right. We show similar performance-cost tradeoff improvements for FrugalGPT for AGNEWS and SCIQ in the Appendix.

avoiding the use of expensive LLMs and improving overall performance. Naturally, a single LLM API is not always correct; FrugalGPT overcomes this by employing a chain of LLM APIs. For example, in the second example shown in Figure 3(a), FrugalGPT identifies that GPT-J's generation may not be reliable and turns to the second LLM in the chain, J1-L, to find the correct answer. Again, GPT-4 provides the wrong answer. FrugalGPT is not perfect, and there remains ample room for cost reduction. For example, in the third example in Figure 3(c), all LLM APIs in the chain give the same answer. However, FrugalGPT is unsure if the first LLMs are correct, resulting in the need to query all LLMs in the chain. How to avoid such cases is an interesting direction of future work.

**Performance Resilience to Data Distribution Shifts.** A common challenge when deploying ML systems in practice is data distribution shifts, i.e., the queries encountered during deployment differ from those in development. To understand the robustness of FrugalGPT against data distribution shifts, we trained FrugalGPT on the original HEADLINES training data and evaluated its performance on four testing datasets with different distributions. Specifically, we created these testing datasets by altering the distribution of labels. For instance, in Variant 1, the label distribution is 33% (up), 17% (down), 17% (none), and 33% (neutral). Conversely, the original dataset's label distribution is balanced (25% for each label). Details can be found in Table 4 in the Appendix. As depicted in Figure 4(a), the performance of both FrugalGPT and GPT-4 remains relatively consistent across different data distributions. Interestingly, while using only 10% of GPT-4's cost, FrugalGPT often delivers similar or superior performance compared to GPT-4 under several testing data distributions.

**Effects of Scorer Functions.** The scorer plays a crucial role in FrugalGPT. Therefore, it is essential to study how the scorer's quality impacts FrugalGPT's performance. In this regard, we focused

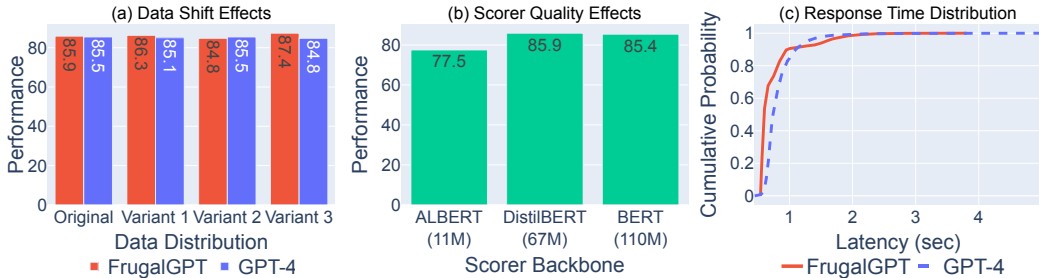

Figure 4: Ablation study of FrugalGPT with a budget of $10\%$ of GPT-4 on the HEADLINES dataset. (a) Effects of data distribution shifts. Each variant corresponds to one label shift instance. Details can be found in the appendix. (b) Effects of scorer quality. (c) Latency (response time) distribution. Overall, the performance of FrugalGPT remains relatively consistent under various testing data distributions different from its training data. As expected, a small and low-quality scorer, such as ALBERT, leads to limited performance, while larger and higher-quality scorers (DistilBERT and BERT) yield better performance. FrugalGPT calls cheaper and faster LLMs on most queries, resulting in shorter response times than GPT-4.

on three backbones for the scorer with varying numbers of parameters: ALBERT (11M), Distil-BERT (67M), and BERT (110M). We trained the scorer on the HEADLINES dataset using different backbone models and compared the performance of the resulting FrugalGPT, with a budget of $10\%$ of GPT-4. As illustrated in Figure 4(b), a low-quality scorer (such as ALBERT) indeed leads to limited performance, as expected. Conversely, larger scorers with better quality, such as DistilBERT and BERT, offer higher performance.

**Improved Latency.** The increasing size of LLMs often correlates with better performance but at the expense of longer response times. Here, we compare the response times of FrugalGPT and GPT-4. Specifically, we set the budget of FrugalGPT to be $10\%$ of GPT-4's cost and compared their performance on the HEADLINES dataset. Overall, we observe that FrugalGPT is often much faster than GPT-4. For instance, 90% of the queries can be answered within 0.9 seconds by FrugalGPT, but more than 1.1 seconds by GPT-4, as shown in Figure 4(c). This is because FrugalGPT learns to call cheaper and faster LLMs for many queries, only invoking the expensive and slow GPT-4 when necessary. Although not explicitly optimized for latency, FrugalGPT inherently provides shorter response times for most queries.

## 5 DISCUSSIONS, LIMITATIONS, AND FUTURE PROSPECTS

The substantial cost of employing LLMs in real-world scenarios presents a considerable barrier to their widespread usage. In this paper, we introduce FrugalGPT, our approach towards resolving this challenge. Our empirical findings show that FrugalGPT can reduce costs by up to 98% while preserving the performance of cutting-edge LLMs.

FrugalGPT lays the groundwork for optimizing task performance with LLM APIs under budget constraints; however, it has some limitations. To train FrugalGPT, we need some labeled examples and additional computational resources. We view this as a one-time upfront cost, which is beneficial when the final query dataset is larger than the data used to train the cascade. There are also other promising strategies for cost saving, such as speeding up attention computation itself and sparsifying LM, that we do not explore here due to limited space. Given the rapid development of LLM, this paper is not meant to be comprehensive, but to lay a foundation for this important research agenda.

There are many related directions for future exploration. While FrugalGPT concentrates on balancing performance and cost, real-world applications call for the evaluation of other critical factors, including latency, fairness, privacy, and environmental impact. Incorporating these elements into optimization methodologies while maintaining performance and cost-effectiveness is an important avenue for future research. Furthermore, utilizing LLMs in risk-critical applications necessitates the careful quantification of uncertainty in LLM-generated outputs. As the field progresses, addressing the environmental ramifications of training and deploying LLMs demands a joint effort from LLM users and API providers.

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
