## A    DISCUSSIONS ON OTHER STRATEGIES

Now we present our vision on two other strategies to use LLM APIs within a budget, namely, *prompt adaptation*, and *LLM approximation*. We give a few examples in Figure 5.

**Strategy 1: Prompt adaptation.**    The cost of an LLM query increases linearly with the size of the prompt. Consequently, a logical approach to reduce the cost of using LLM APIs involves decreasing the prompt's size, a process we refer to as prompt adaptation. *Prompt selection* (as illustrated in Figure 5 (a)) is a natural example of prompt adaptation: rather than employing a prompt containing numerous examples that demonstrate how to perform a task, one can retain a small subset of examples in the prompt. This results in a smaller prompt and subsequently lower cost. An intriguing challenge of prompt selection lies in determining which examples to maintain for various queries without compromising task performance.

An additional instantiation is *query concatenation* (Figure 5 (b)). It is important to note that processing queries individually necessitates sending the same prompt to an LLM API multiple times. Therefore, the fundamental concept of query concatenation involves sending the prompt only once to the LLM API while allowing it to address multiple queries, thereby preventing redundant prompt processing. To accomplish this, several queries must be concatenated into a single query, and the prompt must explicitly request the LLM API to process multiple queries. For instance, to handle two queries using one prompt, the examples presented in the prompt can include both queries followed by their corresponding answers.

**Strategy 2: LLM approximation.**    The concept of *LLM approximation* is quite simple: if an LLM API is too costly to utilize, one can approximate it using more affordable models or infrastructures. One example is the *completion cache*: as depicted in Figure 5 (c), the fundamental idea involves storing the response locally in a cache (e.g., a database) when submitting a query to an LLM API. To process a new query, we first verify if a similar query has been previously answered. If so, the response is retrieved from the cache. An LLM API is invoked only if no similar query is discovered in the cache. The completion cache provides substantial cost savings when similar queries are frequently posed. For instance, consider a search engine powered by an LLM API. If numerous users search for the same or similar keywords simultaneously, the completion cache facilitates answering all their queries by invoking the LLM only once.

Another example of LLM approximation is *model fine-tuning*. As shown in Figure 5(d), this process consists of three steps: first, collect a powerful but expensive LLM API's responses to a few queries; second, use the responses to fine-tune a smaller and more affordable AI model; and finally, employ the fine-tuned model for new queries. In addition to cost savings, the fine-tuned model often does not require lengthy prompts, thus providing latency improvements as a byproduct.

**Compositions.**    Combining approaches within and across different strategies can lead to further cost reduction and performance enhancement. For instance, *joint prompt and LLM selection* is a composition of prompt selection and LLM cascade: for a given query, it searches for the smallest prompt and most affordable LLM that achieves satisfactory task performance. Another example is to search across both existing LLM APIs and fine-tuned models. Note that the composition of different approaches also increases the computational costs for training. Consequently, this paves the way for investigating trade-offs between query costs, task performance, and computational costs.

## B    EXPERIMENT SETUPS AND EXTRA RESULTS

We provide more details on the experiment setups and extra empirical results here.

### B.1    TASKS AND DATASETS

We have evaluated FrugalGPT on five different datasets, ranging from domain-specific classification tasks to general-purpose question answering. The details are summarized in Table 3.Specifically, HEADLINES Sinha & Khandait (2021) is a financial news dataset where the goal is to determine the gold price trend (up, down, neutral, or none) by reading financial news titles. This is especially useful for filtering relevant news in financial markets. OVERRULING Zheng et al. (2021)

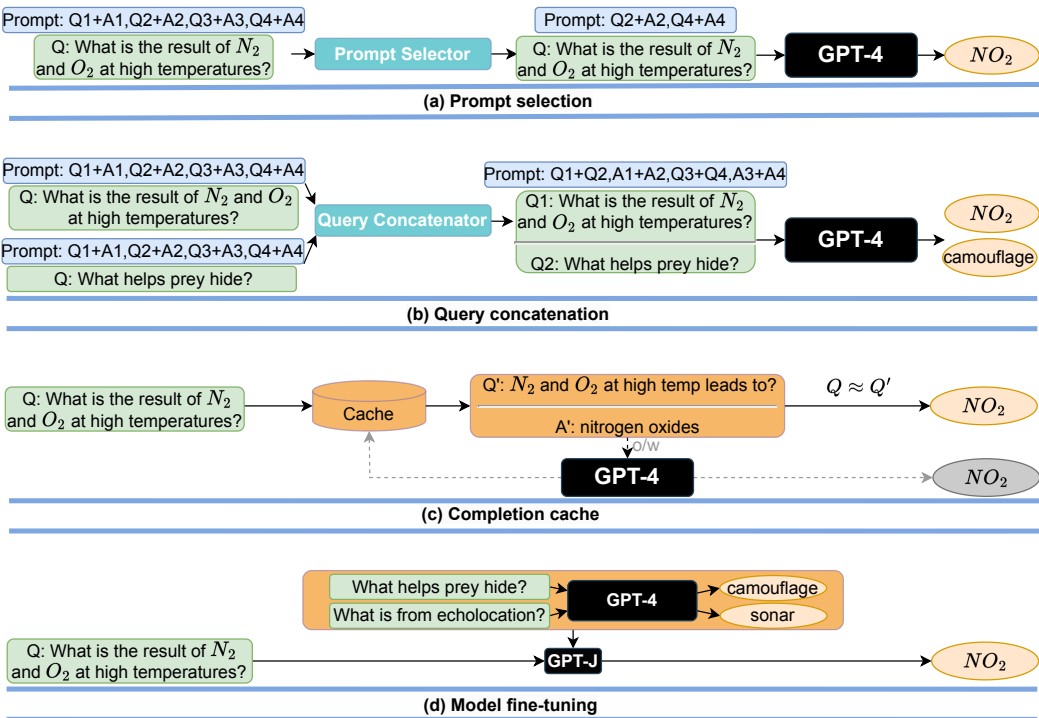

Figure 5: Illustrations of additional cost-saving strategies. (a) Prompt selection uses a subset of in-context examples as the prompt to reduce the size of the prompt. (b) Query concatenation aggregates multiple queries to share prompts. (c) Completion cache stores and reuses an LLM API's response when a similar query is asked. (d) Model fine-tuning uses expensive LLMs' responses to fine-tune cheap LLMs.

Table 3: Summary of datasets used in the FrugalGPT LLM cascade experiments.

| Dataset | Domain | Size | #Examples in the prompt |
|---------|--------|------|-------------------------|
| HEADLINES | Finance | 10000 | 8 |
| OVERRULING | Law | 2400 | 5 |
| COQA | Multi-domain | 7982 | 2 |
| AGNEWS | Journalism | 7600 | 8 |
| SCIQ | Science | 11680 | 8 |

is a legal document dataset where the goal is to determine whether a given sentence is an over-ruling, i.e., rejecting previous legal cases. COQA Reddy et al. (2019) is a multi-domain reading comprehension dataset developed in a conversational setting, which we have adapted as a direct query-answering task. AGNEWS Zhang et al. (2015) is a news dataset. The task is to classify each news into one of four categories (business, world, sports, and sci/tech). SCIQ Welbl et al. (2017) is a scientific question-answering dataset. Given a short paragraph, the target is to answer a question about physics, chemistry, and biology.

We reformat all datasets so that the LLMs process each data point as a few-shot learning problem. In particular, we append (i) a short description of the task alone with (ii) a few examples in the prompt to each data point and then feed it to the LLMs. The details of the prompt prefix is given in the following subsection. For AGNEWS, we randomly select the examples from their original training partitions. There are no official train-eval partitions for HEADLINES and OVERRULING, so we randomly select a few in-context examples from the entire datasets, and evaluate the performance of commercial APIs and FrugalGPT on the remaining data points. AGNEWS, COQA, and SCIQ all offer their official training and evaluation partitions. Due to budget limits, in-context examples

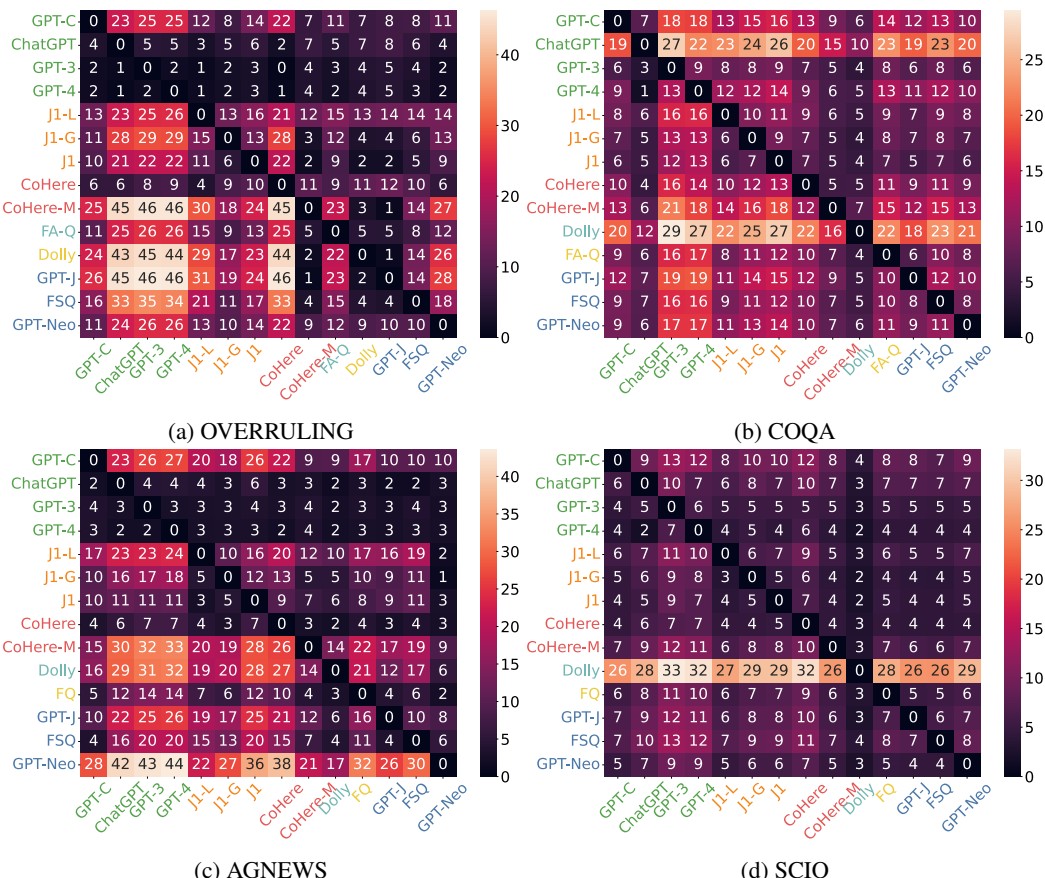

Figure 6: Maximum performance improvement (MPI) of each pair of LLMs. (a), (b), (c) and (d) correspond to the four datasets, separately. One entry indicates the percent of cases that the LLM on its row is wrong but the LLM on its column gives the right answer. Overall, we observe that cheap LLMs can be complementary to expensive ones quite often. For example, for about 5% of the data, GPT-4 makes a mistake but GPJ-J gives the right answer on OVERRULING.

are randomly selected from the larger partitions (often the training partition) and the evaluation is performed on the evaluation partitions only. The training partition of SCIQ is relatively smaller so we evaluate the performance on the larger partition and select the in-context examples from the smaller partition.

## B.2 ADDITIONAL EVALUATIONS

Here we present additional empricial results, including the LLM diversity measurements and the performance-cost trade-off for all datasets.

**LLM Diversity.** In Section 4, the study of the MPI for HEADLINES reveals a large potential for performance improvements over the best individual LLM. Does this generalize to other datasets as well? Here, MPI between each pair of LLM APIs for the remaining 4 datasets is displayed in Figure 6. Overall, we observe a similar phenomenon. For instance, GPT-J, can enhance GPT-4's performance by up to 5% on the OVERRULING dataset. On the COQA dataset, there are 13% of data points where GPT-4 makes an error, but GPT-3 provides the correct answer. Although these improvement upper bounds may not always be attainable, they do demonstrate the possibility of utilizing more affordable services to achieve better performance.

**Additional Performance-Cost Trade-offs.** The performance and cost trade-offs attained by Fru-galGPT on AGNEWS and SCIQ are presented in Figure 7. The overall trends are the same as the

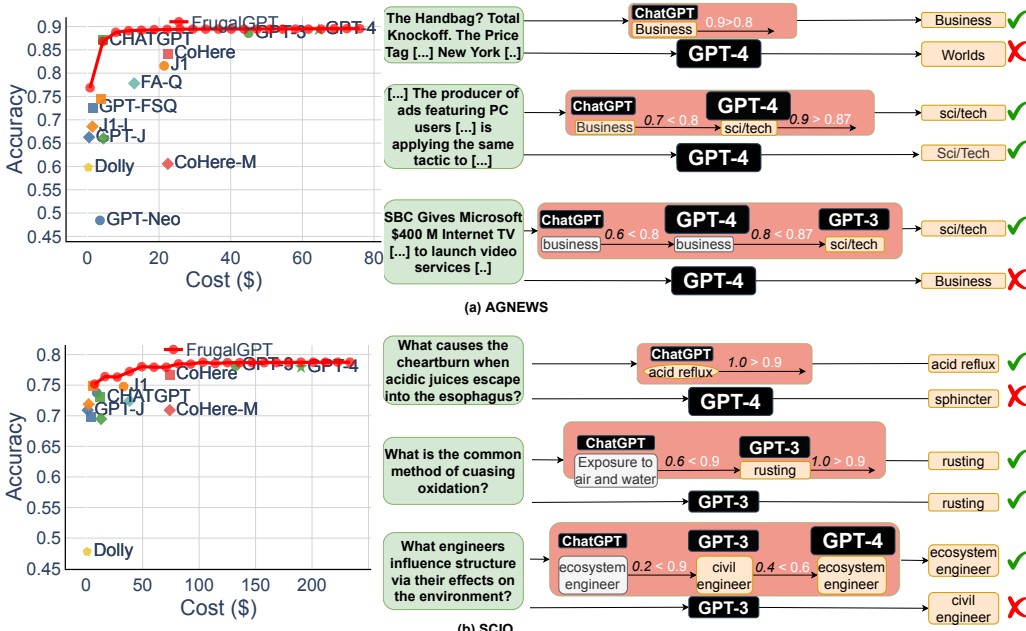

Figure 7: Accuracy and cost tradeoffs achieved by FrugalGPT on AGNEWS and SCIQ. Overall, FrugalGPT achieves the same performance of the best individual LLM API (e.g., GPT-4) with orders of magnitudes smaller cost. Examples of LLM cascade for each dataset are shown on the right.

Table 4: Synthesized data distribution shifts for the ablation study. The label distribution of the original evaluation data is balanced. The three variants are all generated by varying the marginal label distribution. The table shows the fraction of different labels for each variant.

| Eval Data Instance | Label Distribution | | | |
|---|---|---|---|---|
| | Up | Down | None | Neutral |
| Original | 0.25 | 0.25 | 0.25 | 0.25 |
| Variant 1 | 0.33 | 0.17 | 0.17 | 0.33 |
| Variant 2 | 0.40 | 0.20 | 0.20 | 0.20 |
| Variant 3 | 0.50 | 0.17 | 0.17 | 0.17 |

other datasets shown in Figure 3: FrugalGPT offers flexible budget-aware strategies as well as significant cost reduction when matching the performance. The examples (on the right panel of the figure) provide a few dataset-specific insights. For example, GPT-4 incorrectly classified the first example in AGNEWS as "Worlds" while the correct answer is "business" as produced by Frugal-GPT. This is probably because there are a few city names (e.g., New York) in the original news. The first example in the SCIQ dataset is also quite interesting. GPT-4's answer, "sphincter", was incorrect but reasonable: sphincter does offer the power for many movements in the stomach. However, it is not as precise as "acid reflux", the correct answer produced by FrugalGPT. This suggests that GPT-4 may have difficulty distinguishing high-level concepts from low-level concepts within certain contexts. FrugalGPT, on the other hand, learns to identify this and adapt to the most trustful LLM as needed.

## B.3 DATA SHIFT SYNTHESIS

To study the robustness to data distribution shifts, we have synthesized three data distributions different from the original data distribution (Figure 4).Note that the original data distribution can be

decomposed as $\Pr[x, y] = \Pr[y] \Pr[x|y]$. To generate the variants, we fixed $\Pr[y|x]$ and vary the marginal label distribution. The corresponding label distribution is shown in Table 4.

For completeness, we also conducted an additional distribution shift study. Specifically, we asked ChatGPT to rephrase each question without changing its meaning. On the HEADLINES dataset, we observe that FrugalGPT is quite robust to such shifts: GPT-4's achieves 84% overall performance on the shifted dataset, and FrugalGPT can reach the same accuracy with 33% of the cost.

### B.4 COMPARISONS WITH OUTPUT ENSEMBLE

We conducted an additional experiment for the model ensembles. In particular, we query ChatGPT 3 times with different temperatures (0, 0.1, 0.2) and then take the mode of the generation to be the final output. As shown in Table 5, sampling multiple times leads to marginal performance gains on the HEADLINES dataset. Yet, the gain is relatively smaller than FrugalGPT, whose accuracy can be 87% at the same cost.

Table 5: Comarison with ChatGPT ensemble on the HEADLINES dataset. Overall, FrugalGPT achieves much better performance with the same cost of the ensemble.

| ChatGPT (1 output) | ChatGPT ensemble (3 outputs) | FrugalGPT |
|---|---|---|
| 81.00% | 83.80% | 87% |

### B.5 EFFECTS OF TRAINING DATASET SIZE

We also conducted an additional study on training set size effects on the HEADLINES dataset. Specifically, we varied the data used to train FrugalGPT from 500 to 5000, and measured the accuracy of the FrugalGPT LLM cascade on the test set with 5000 samples. As shown in Table 6, FrugalGPT's performance was consistently better than GPT-4 (85.5%).

Table 6: Training size effects of FrugalGPT on the HEADLINES dataset.

| Training set size | Accuracy |
|---|---|
| 500 | 85.9 |
| 1000 | 86.8 |
| 2000 | 86.8 |
| 3000 | 86.8 |
| 4000 | 87.2 |
| 5000 | 87 |

## C PROMPT DETAILS

Here we provide the prompts used for each task. In a nutshell, we use few-shot prompting for robust performance.

### C.1 PROMPT FOR HEADLINES

> Please determine the price direction (up, down, neutral, or none) in the following news headlines.
> Q: december gold down $1 at $749 an ounce on nymex
> A: down
> Q: august gold up $7.60 at $878.80 an ounce on nymex
> A: up
> Q: commodity outlook: gold may find it tough to top 30,920 level

A: none
Q: gold prices at 1-week lows as dollar remains supported
A: neutral
Q: illegal flow of gold to nepal from india across porous border showing upward swing
A: none
Q: gold prices steady in early asia trade with focus on hong kong
A: neutral
Q: gold adds 0.7% to trade at record $1,601.50/oz
A: up
Q: gold loses sheen on muted demand, silver also eases
A: down

## C.2 PROMPT FOR OVERRULING

Context: because jones/walker relates only to sufficiency of the evidence, we hereby disavow the language holding otherwise in sandoval.
Question: Is it overruling?
Answer: Yes
Context: insofar as the givens case holds contrary to our original opinion herein or to the rule expressed in the carr case and the cases there cited, it is expressly overruled.
Question: Is it overruling?
Answer: Yes
Context: the court also specifically ordered defendant to "be subject to all administrative or judicial enforcement remedies available to the plaintiff as prescribed by state and federal law in a title iv-d case[.]"
Question: Is it overruling?
Answer: No
Context: according to napa auto parts, the straws drove the vehicle "for approximately six [] weeks and [] for between 500 to 600 miles prior to the accident with no incidents."
Question: Is it overruling?
Answer: No
Context: accordingly, we answer the certified question in the affirmative, disapprove deruyter, and approve the decision of the court below.
Question: Is it overruling?
Answer: Yes

## C.3 PROMPT FOR COQA

Context: Michigan () is a state in the Great Lakes and Midwestern regions of the United States. The state's name, Michigan, is of French origins (form of the Ojibwe word) "mishigamaa", meaning "large water" or "large lake". Michigan is the tenth most populous of the 50 United States, with the 11th most extensive total area, and the largest state by total area east of the Mississippi River. Michigan's capital is Lansing, and its largest city is Detroit. Michigan is the only state to consist of two peninsulas. The Lower Peninsula, to which the name Michigan was originally applied, is often noted to be shaped like a mitten. The Upper Peninsula (often referred to as "the U.P.") is separated from the Lower Peninsula by the Straits of Mackinac, a channel that joins Lake Huron to Lake Michigan. The two peninsulas are connected by the Mackinac Bridge. The state has the longest freshwater coastline of any political subdivision in the world, being bounded by four of the five Great Lakes, plus Lake Saint Clair. As a result, it is one of the leading U.S. states for recreational boating. Michigan also has 64,980 inland lakes and ponds. A person in the state is never more than from a natural water source or more than from a Great Lakes shoreline.
Question: what separates the two peninsulas?
Answer: the Straits of Mackinac,
Context: Harry had a very small farm. He only had one cow but dreamed about having a large farm. He once asked his father Bill, "I'd like to have that land over there. How can I get it?" His father encouraged him to go and talk to the landowner to see how they could get the land. Harry said. "But we don't have enough money." His father said, "Don't worry. Go and talk to him." Several years passed. Harry had not only the land, but also several hundred cows. He had a happy life with his wife. Later, Harry's wife, Sarah, had a dream. "I want to build the biggest farm in the world." She said. They called their friend Manuel about this task. Three days later Manuel had a plan for the whole project. Then they asked, "How much will it cost?" Manuel said they needed a lot of money. "Nobody will lend us so much money to build a farm," they thought. But the manager of the bank them and their dream. A few months later, La manuel, the biggest farm in the world, was opened.
Question: Who was the farmer?
Answer: Harry

## C.4 PROMPT FOR AGNEWS

Please answer which category (World, Sports, Business or Sci/Tech) a provided news falls into.
Q: Five-year ban for Blackburn fan One of the two Blackburn Rovers Football Club fans charged with public disorder for racially abusing Dwight Yorke has been handed a five-year ban.
A: Sports
Q: Major software pirates caught A multimillion-euro software piracy ring has been broken following synchronized raids in Athens and London yesterday, Attica police said.
A: Sci/Tech
Q: Loews to Buy Entergy-Koch Pipeline NEW YORK (Reuters) - Conglomerate Loews Corp. <A HREF="http://www.investor.reuters.com/FullQuote.aspx?ticker=LTR.Ntarget=/stocks/quickinfo/fullquote">LTR.N</A> agreed to buy an 8,000-mile natural gas pipeline system from Entergy-Koch LP for $1.14 billion on Monday, in a bid to cash in on rising U.S. demand for natural gas.
A: Business
Q: Texas A amp;M Quarterback Finds Groove Once Again Reggie McNeal switched his jersey number in the off-season, trading No. 16 for No. 1 in a salute to a departed teammate. McNeal has become the
A: Sports

Q: UPDATE 2-Rugby-Australia edge out England in Twickenham thriller Australia showed all their famed resilience to withstand a fierce fightback by England and beat the world champions 21-19 in a thunderous World Cup final repeat on Saturday.
A: Sports
Q: Ed Hardin: Bowl situation not so Peachy CHAPEL HILL; They came down from the hills Saturday, down from the hot springs and natural bridges of Virginia, deep into the heart of ACC darkness.
A: Sports
Q: Democrat Seeks Probe of Bush Aides' Travel (AP) AP. The chairwoman of the House Democrats' homeland security task force is asking Congress' independent auditors to examine travel by senior Bush administration officials in light of recent trips to hotly contested states in the 2004 presidential election.
A: World
Q: Chelsea: #39;No fear #39; factor in Europe IF a football team could be entered in that famous television programme Fear Factor, then Jose Mourinho would register Chelsea. Because if Chelsea are going to reach the final of the Champions League for the
A: Sports

## C.5 PROMPT FOR SCIQ

Please answer the following questions concisely.
Context: inside these cells, glucose is immediately converted into glucose-6-phosphate. By doing this, a concentration gradient is established where glucose levels are higher in the blood than in the cells. This allows for glucose to continue moving from the blood to the cells where it is needed. Insulin also stimulates the storage of glucose as glycogen in the liver and muscle cells where it can be used for later energy needs of the body. Insulin also promotes the synthesis of protein in muscle. As you will see, muscle protein can be catabolized and used as fuel in times of starvation. If energy is exerted shortly after eating, the dietary fats and sugars that were just ingested will be processed and used immediately for energy. If not, the excess glucose is stored as glycogen in the liver and muscle cells, or as fat in adipose tissue; excess dietary fat is also stored as triglycerides in adipose tissues. Figure 24.21 summarizes the metabolic processes occurring in the body during the absorptive state.
Question: Excess dietary fat is stored as triglycerides in the body. what type of tissue is used to store the triglycerides?
Answer: adipose
Context: trees that lose their leaves once a year.
Question: What are trees that lose their leaves during winter called?
Answer: deciduous
Context: Primary Vesicles As the anterior end of the neural tube starts to develop into the brain, it undergoes a couple of enlargements; the result is the production of sac-like vesicles. Similar to a child's balloon animal, the long, straight neural tube begins to take on a new shape. Three vesicles form at the first stage, which are called primary vesicles. These vesicles are given names that are based on Greek words, the main root word being enkephalon, which means "brain" (en- = "inside"; kephalon = "head"). The prefix to each generally corresponds to its position along the length of the developing nervous system. The prosencephalon (pros- = "in front") is the forward-most vesicle, and the term can be loosely translated to mean forebrain. The mesencephalon (mes- = "middle") is the next vesicle, which can be called the midbrain. The third vesicle at this stage is the rhombencephalon. The first part of this word is also the root of the word rhombus, which is a geometrical figure with four sides of equal length (a square is a rhombus with 90° angles). Whereas prosencephalon and mesencephalon translate into the English words forebrain and midbrain, there is not a word for "four-sided-figure-brain. " However, the third vesicle can be called the hindbrain. One way of thinking about how the brain is arranged is to use these three regions—forebrain, midbrain, and hindbrain—which are based on the primary vesicle stage of development (Figure 13.3a).

Question: As the anterior end of the neural tube starts to develop into the brain, it undergoes a couple of enlargements; the result is the production of these?
Answer: sac-like vesicles
Context: The immune response mainly involves the lymphatic system. The lymphatic system is a major part of the immune system. It produces leukocytes called lymphocytes. Lymphocytes are the key cells involved in the immune response. They recognize and help destroy particular pathogens in body fluids and cells. They also destroy certain cancer cells.
Question: What system of the body is most involved in the immune response?
Answer: lymphatic system
Context: Carbohydrates are organic molecules that consist of carbon, hydrogen, and oxygen. They are made up of repeating units called saccharides. They provide cells with energy, store energy, and form structural tissues.
Question: What are organic molecules that consist of carbon, hydrogen, and oxygen called?
Answer: carbohydrates
Context: Materials that are poor conductors of thermal energy are called thermal insulators. Gases such as air and materials such as plastic and wood are thermal insulators.
Question: What are materials that cannot conduct thermal energy efficiently known as?
Answer: thermal insulators
Context:
Question: In our wildflower population, the pool of what remains constant from one generation to the next?
Answer: genes
Context: The relative sizes of the atoms show several trends with regard to the structure of the periodic table. Atoms become larger going down a column and smaller going across a period.
Question: The relative sizes of the atoms show several trends with regard to what visual method of organization?
Answer: periodic table

# D    ADDITIONAL DISCUSSIONS

**Design space constraints due to black-box access.**    Note that the design space of FrugalGPT is restricted by the black-box access to LLM APIs. In this paper, we only assume access to each LLM's response to a user query, which holds for all LLM APIs considered in this paper. Relaxation of this assumption gives FrugalGPT more information and thus leads to potentially more effective design. For example, if the token likelihood value/vector is available, then one may use it to quantify an LLM's own uncertainty about its generation for a more effective scorer.

**Evaluation scopes.**    Tasks with short outputs often come with ground-truth labels that enable objective evaluations. Many LLM benchmarks and evaluations rely on short outputs or multiple choice questions. Quality evaluation for long generations, on the other hand, often involves LLM as a judger, with potential biases and subjectiveness. Thus, we focus on the former ones to ensure the evaluation is objective and matches human judgments.

**Justification for the chosen datasets.** We chose the datasets because they represented an important subset of real-world applications. For example, the HEADLINES and OVERRULING datasets represent the task of text classification and tagging, which is listed as an important application of GPT-4 by OPENAI (see, e.g., https://platform.openai.com/examples/default-tweet-classifierandhttps://platform.openai.com/examples/default-review-classifier). The SCIQ dataset measures the scientific question answering ability, which is key for QA applications such as the Khan academy pilot program (see https://openai.com/gpt-4).

**Challenges for training the scorer.**    There is no numerical label for quality, and thus we cannot use the standard regression training paradigm directly. Instead, we trained a model to predict cor-

rectness, which is binary and can be curated from the ground-truth labels. Then we used the model's confidence as the judge's quality estimation.