# OpenReview forum: "FrugalGPT: How to Use Large Language Models While Reducing Cost and Improving Performance"
_ICLR.cc/2024/Conference — Submitted to ICLR 2024_

### Official Review · Reviewer_dfGd · 2023-10-31

**Soundness:** 4 excellent
**Presentation:** 3 good
**Contribution:** 3 good
**Rating:** 8
**Confidence:** 5

**Summary:**

The paper presents and approach to significantly reduce the cost of querying LLMs by using a combination of LLMs of multiple sizes and deciding when and which model to query based on an auxiliary scorer model. The quality of scorer model seems key to achieving good performance, but the paper shows that even if much smaller sized backbone models for the scorer, the model is able to perform pretty good.

The paper presents multiple experiments to demonstrate the strength of the method showing that apart from the cost reduction, having such a router can also improve the overall quality of the responses given the routers ability to understand which LLM is suitable for which type of task.

**Strengths:**

- Good contribution to key problems on affordability and environmental challenges from LLMs.
- Pretty significant cost reduction while using very small scorer models to route the queries.
- The setup is resilient to a good extent to data distribution shift even though it seems like the quality of the scorer model would be significantly dependent on the ability of the model to learn distribution of the queries, which is generally observed to be limited in terms of generalizability for small models.

**Weaknesses:**

- There is a need for an auxiliary model to be trained (scorer).
- The scorer model could significantly determine the gains from the setup.
- There is need for labelled dataset, and clients need to call the auxiliary model for each query, which will often not be on as performant hardware as the LLM APIs and could lead to latency hits.

**Questions:**

None, but feel free to add comments on the weakness section.

---

> ### Author Response · Authors · 2023-11-21
>
> Thank you for your helpful feedback and support for the paper!

---

### Official Review · Reviewer_q9b6 · 2023-10-31

**Soundness:** 3 good
**Presentation:** 3 good
**Contribution:** 4 excellent
**Rating:** 8
**Confidence:** 4

**Summary:**

FrugalGPT proposes a layer on top of competing LLMs designed to minimize costs without hurting performance or latency, by adaptively selecting LLMs and evaluating their answers with smaller models.

**Strengths:**

* This is a simple and brilliant idea, with tremendous potential impact for affordable accessibility of AI, minimized environmental impact, and general efficiency.
* The authors leverage an intuitive observation (answer validation is easier than answer generation) and hierarchy of models to save costs and improve performance together.
* Demonstration of improved latency is an added benefit

**Weaknesses:**

* No justification for the chosen datasets. Do these map to diverse and realistic domains, tasks, and applications? The evaluation selection is fairly important to understand what realistic cost saving ranges are.
* Evaluation datasets are all short outputs: classification or question answering. This is a slightly limited setting for the claims.

**Questions:**

* Are any of the datasets speculated to be in the training set of some of the models? Is there a way to test this?

---

> ### Author Response · Authors · 2023-11-21
> **Thank you for your helpful feedback and support for the paper**
>
> Thank you for your helpful feedback and support for the paper! We answer your questions below.
>
> ***[Justification for the chosen datasets]***: We chose the datasets because they represented an important subset of real-world applications. For example, the HEADLINES and OVERRULING datasets represent the task of text classification and tagging, which is listed as an important application of GPT-4 by OPENAI (see, e.g., https://platform.openai.com/examples/default-tweet-classifier and https://platform.openai.com/examples/default-review-classifier). The SCIQ dataset measures the scientific question answering ability, which is key for QA applications such as the Khan academy pilot program (see https://openai.com/gpt-4). We have also added a discussion on this in the revised draft (see Section D in the appendix).
>
> ***[Evaluations are short outputs]***: Tasks with short outputs often come with ground-truth labels that enable objective evaluations. Many LLM benchmarks and evaluations rely on short outputs or multiple choice questions. Quality evaluation for long generations, on the other hand, often involves LLM as a judger, with potential biases and subjectiveness. Thus, we focus on the former to ensure the evaluation is objective and matches human judgments. We have updated the revised draft to make this clear (see Section D in the Appendix).
>
> ***[Are any datasets in the training set of some models]***: Data contamination is indeed important. Without access to the training datasets and documents of black-box commercial LLM services,  however, it is challenging to definitively detect if a data point has been seen during training.

---

### Official Review · Reviewer_3Fst · 2023-11-01

**Soundness:** 2 fair
**Presentation:** 3 good
**Contribution:** 2 fair
**Rating:** 5
**Confidence:** 3

**Summary:**

The paper proposes FrugalGPT, a pipeline for saving API costs for a given task, by progressively calling more capable APIs and stoping when a judge (verifier) determines the answer is good enough. On 5 different benchmarks, it is shown that FrugalGPT can save costs by 52%-98% depending on the task.

**Strengths:**

* FrugalGPT is a simple and effective method for saving API costs. It has practical utilities.
* The paper is in general well-written.

**Weaknesses:**

* The benefit of FrugalGPT over simple methods such as fine-tuning is unclear.
  * Experiments are mainly conducted on QA and text classification tasks. From my understanding, a baseline of fine-tuning RoBERTa will achieve 90%+ accuracy on AGNews and CoQA.
  * In terms of data efficiency, FrugalGPT also requires a training set to train the judge and the permutation function, and need extra information to compute MPI.
  * From my understanding, calling API is most common when there are only few training examples, or the task is complex. It would be more convincing to show that FrugalGPT works well in these cases.
* Many details are missing in the current version of the paper, see questions below.

**Questions:**

Questions:
* Is FrugalGPT a task-specific or task-agnostic method? In the current experiments, are the parameters (permutation, judge model, threshold) trained separately for each task?
* What's the cost of estimating MPI? Does that mean you need to run all LLMs over all training examples?
* What data is used to train the DistilBERT judge?
* For classification tasks, is it possible to use the DistilBERT judge directly for inference?
* Since the judge model is doing a regression problem, what's the reason of using a 2-d output, and taking the maximum value of the last layer (normalized by softmax)?
* As generation diversity is mentioned as a reason for improving performance, would it be possible to sample outputs from the same LLM multiple times, and ensemble their answers. As shown in https://arxiv.org/abs/2203.11171 (Figure 2) this would also create a cost-performance curve, and the performance grows with more calls. To demonstrate the contribution of FrugalGPT, such curves should also be included for comparison.
* In Figure 1(d), I'm a little confused why the leftmost dot of FrugalGPT is still better than the cheap models. At this budget point, I would assume FrugalGPT makes one single call to one API, and thus the dot of FrugalGPT should overlap with one of the model. Please help me understand this better.

Suggestion:
* Regarding the distribution shift experiments, currently it's mainly done by changing the label distribution in the test set. It would be interesting to study and evaluate how FrugalGPT performs when there is distribution shift on the input text.

---

> ### Author Response · Authors · 2023-11-21
> **Thank you ver much for your time and feedback! We have answered your questions as below (1/2)**
>
> Thank you very much for your time and feedback.
>
> ***[The benefit of FrugalGPT over simple methods such as fine-tuning]***: Fine-tuning requires more data. For example, 127k data points are needed to fine-tune a ROBERTa model on COQA [R1], while only a few hundred samples are needed for FrugalGPT. In addition, FrugalGPT is complementary to fine-tuning an individual LLM since we can wrap FrugalGPT on top of the fine-tuned LLM and other LLMs to reduce cost.
>
> [R1], I. Staliūnaitė et al, Compositional and Lexical Semantics in RoBERTa, BERT and DistilBERT: A Case Study on CoQA, 2020.
>
> ***[Experiments are mainly conducted on QA and text classification tasks]***: These tasks are selected partially because they are relatively easy to evaluate, which is why these datasets are commonly used to evaluate LLM research. In addition, several of our datasets are quite challenging. For example, on the COQA and SCIQ datasets, even GPT-4 is only able to achieve 40%-70% accuracy.
>
> ***[FrugalGPT requires a training set to train the judge and the permutation function, and need extra information to compute MPI]***: FrugalGPT is designed for applications involving a large number of LLM queries. Consider, for example, Yabble (https://openai.com/customer-stories/yabble), which analyzes customer posts for sentiment,  or, Jasper (https://www.jasper.ai/the-prompt/gpt4-future-of-gen-ai-business), which offers chatbots for varying domains. Here, as in many other industry applications of LLM, many tens of thousands of new queries per day are expected. In these cases, the cost of tuning FrugalGPT on a few thousand examples is well worth the reduction in cost when applying it to millions over time. We also conducted an additional study on the effects of FrugalGPT training set size in the HEADLINES dataset.  Specifically, we varied the data used to train FrugalGPT from 500 to 5000, and measured the accuracy of the FrugalGPT LLM cascade on the test set with 5000 samples. As shown in the following table, FrugalGPT’s performance was consistently better than GPT-4 (85.5%). We have added this in the revised paper (see Section B.5 in the Appendix).
>
> | Training set size            | Accuracy |
> | :----------------: | :------: |
> | 500        |   85.9   |
> | 1000           |  86.8  |
> | 2000   | 86.8 |
> | 3000   | 86.8 |
> | 4000   | 87.2 |
> | 5000   | 87.0 |
>
>
> ***[From my understanding, calling API is most common when there are only few training examples, or the task is complex]***: Many enterprises are interested in using LLMs on a large number of relatively repetitive tasks. For example, financial companies such as Stripe are using LLMs to perform large scale fraud detection (https://www.techcircle.in/2023/03/16/11-companies-using-gpt-4-in-consumer-products), and Yabble analyzes customer posts for sentiment (https://openai.com/customer-stories/yabble), which is similar to our evaluation datasets AGNEWS. Insurance companies are also using LLMs to process millions of claims. Also, FrugalGPT requires fewer labeled data points than fine tuning a model for the task would (since it only needs to train a router that can still use the big model).
>
> ***[Is FrugalGPT a task-specific or task-agnostic method]***: The FrugalGPT paradigm is task-agnostic, but different tasks do require additional training for better performance. For example, our experiments show the cost savings due to FrugalGPT in diverse tasks including price prediction, reading comprehension and news classification.
>
> ***[What's the cost of estimating MPI]***: MPI was estimated based on each API’s performance on the training partitions. FrugalGPT was designed for applications requiring numerous and continuous queries to LLMs. In such applications, the cost of estimating MPI is often small (less than 5%) compared to processing queries during deployment.
>
> ***[What data is used to train the DistilBERT judge]***: The judge was trained on the training partition of each evaluation dataset. The judge training is data efficient: a few hundred examples are enough to obtain a high quality judge.
>
> ***[Is it possible to use the DistilBERT judge directly for inference for classification tasks]: Yes. One can use the judge to process a query with each possible label and pick the label with the maximum score returned by the judge. However, this performance is much worse than FrugalGPT. For example, on the HEADLINES dataset, directly using the judge for inference gives an accuracy of 60% performance compared to the FrugalGPT’s accuracy of 87%. This is because judging is easier than prediction. For example, for a given query, the judge may assign low scores to all possible labels. In this case, FrugalGPT would defer the query to the next LLM, which may give a high-quality answer. Since all assigned scores are low, directly using the judge for labeling can give poor predictions.

---

> ### Author Response · Authors · 2023-11-21
> **Thank you ver much for your time and feedback! We have answered your questions as below (2/2)**
>
> ***[What’s the reason of a 2-d output for training the judge as a regression problem]***: This is because there is no numerical label for quality, and thus we cannot use the standard regression training paradigm directly. Instead, we train a model to predict correctness, which is binary and can be curated from the ground-truth labels. Then we use the model’s confidence as the judge’s quality estimation. We have added a detailed discussion in the revision (Section D in the appendix).
>
> ***[Compare with sampling outputs from the same LLM multiple times, and ensembling their answers]***: We have conducted an additional experiment for the model ensembles. In particular, we query ChatGPT 3 times with different temperatures (0, 0.1, 0.2) and then take the mode of the generation to be the final output. As shown in Table 5 in the appendix, sampling multiple times leads to marginal performance gains on the HEADLINES dataset. Yet, the gain is relatively smaller than FrugalGPT, whose accuracy can be 87% at the same cost.  We have added this in the appendix (Section B) too.
>
> ***[Why the leftmost dot of FrugalGPT is still better than the cheap models in Figure 1(d)]***: This is because the leftmost dot of FrugalGPT actually used a bit more budget than the cheapest model. In fact, it routed 2% of queries to ChatGPT. This leads to an accuracy gain of 0.7% (=78.7%-78%).
>
> ***[It would be interesting to study and evaluate how FrugalGPT performs when there is distribution shift on the input text]***: We conducted new experiments for distribution shift on the input text. Specifically, we asked ChatGPT to rephrase each question without changing its  meaning. On the HEADLINES dataset, we observe that FrugalGPT is quite robust to such shifts. For example, GPT-4 achieves 84% overall performance on the shifted dataset, and FrugalGPT can reach the same accuracy with only 33% of the cost. We have added this analysis in the revised draft (see Section B.3 in the Appendix).
>
> Thank you again for your time! We hope you would consider increasing your score if we have answered your questions. Please let us know if you have additional questions and we are happy to follow up. Thanks!

---

### Official Review · Reviewer_QsiX · 2023-11-10

**Soundness:** 2 fair
**Presentation:** 4 excellent
**Contribution:** 4 excellent
**Rating:** 6
**Confidence:** 5

**Summary:**

Authors propose a technique called FrugalGPT which selects which sequence of model APIs to call in order to reduce costs.  FrugalGPT has 3 different trainable components: router, answer-scorer and user budget. In experiments, they find that there is upto a 98% cost reduction, and has similar or slightly better results than using the strongest API used for experiments. In addition, they investigate the effect of data distribution, scorer quality changes and the incidental latency improvements that come along with cost.

**Strengths:**

- The authors spend time motivating a novel problem faced by practitioners in a clear and engaging manner.
- The paper is in general well-written and I did not have trouble following the proposed method.
- The results are compelling and exciting.
- The analysis done is useful to practitioners (latency improvements, distribution shift, scorer quality)

**Weaknesses:**

The main weakness of the paper is the lack of analysis on what makes the method works. This method is relatively complex, with three different components, and there are several natural ablations that have not been done. While it is understandable that there are API costs to contend with, even using open source models like Llama with synthetically assigned costs for ablations could have been done. However, I still believe that the paper should be accepted.

Given that all related papers that are not speculative decoding are too recent for the authors to take into account (around/after ICLR deadline), it is fine not to have external baselines to compare to (speculative decoding provides better latency, but does not have the same cost improvements, afaik). However, some ablations on the method would be useful:
- How much does the search space pruning help in practice, versus just ordering the APIs in increasing order of cost? (vs just anecdotes shown in the experimental section)
- How much changing the threshold T effected results? Is the method very sensitive to T? This would be useful to practitioners.

Some of the design space is indeed constrained by only having access to the output (see comments on scorer in Questions section). I believe writing this in a discussion somewhere in the draft would be helpful in answering questions a reader would have on a first reading.

**Questions:**

- Why is a scorer necessary? Given closed APIs don't give probability/logit outputs generally, this is not an option. However, when there is an option, is a simpler proxy like confidence or probability thresholding enough? (like using a much smaller learned calibration module at the query level Eg. https://arxiv.org/abs/2207.07411 https://arxiv.org/abs/2207.05221. A logit based score is more useful for routing at the token level).
- While interesting, the latency improvments are modest relative to what is possible by combining the use of expensive/larger and cheaper/smaller models (speculative decoding, and https://arxiv.org/abs/2310.12126 get a 2-3x improvement in latency). Do you believe that additionally optimizing for this can result in similar improvements to latency and an improvement in cost (ie, either replacing cost budget with latency budget, or combining the two)? Or is this a limitation of relying on closed source models?

---

> ### Author Response · Authors · 2023-11-21
> **Thank you for your helpful feedback and support of the paper!**
>
> Thank you for your helpful feedback and support of the paper! We answer your questions as follows.
>
> ***[How much does the search space pruning help in practice, versus just ordering the APIs in increasing order of cost?]***: Pruning does help identify the most effective set of LLMs to query. As an ablation study, we compare the performance of FrugalGPT and that of using three APIs in increasing order of cost: GPT-J, GPT-Neo, and GPT-4. We chose these three APIs because they roughly lie on the surface of the convex hull of all APIs in Figure 1. As shown below, pruning (FrugalGPT) brings additional 1-2% performance improvements compared to using a fixed ordering of APIs in increasing cost.
>
> | Cost (fraction of GPT-4) | FrugalGPT | FrugalGPT with fixed API order|
> | :----------------: | :------: | :------: |
> | 10%  | 86.8 |85.1 |
> | 20%  | 86.9 |85.6 |
> | 100%  | 87.2 | 85.8|
>
> ***[How much changing the threshold T affected results? Is the method very sensitive to T?]***: The threshold T does affect the performance, as it controls when a query is routed to an API. Fortunately, practitioners do not need to tune it themselves: the threshold is learned automatically during FrugalGPT’s training process (the optimization problem on the bottom of page 4).
>
> ***[Writing a discussion on the constraints of the design space]***: Thank you for the helpful suggestion! We have added a short discussion in the revised draft (see Section D in the appendix).
>
>
>
> ***[If APIs give probability/logit outputs, is a scorer still necessary?]***: The scorer is a flexible module to measure the quality of a generation; it does not necessarily need to be the particular form we used. If the APIs give probability/logit outputs that truly capture the quality of their generations, then leveraging these signals can certainly simplify the scorer design. However, in practice, the API’s probability and logit are often not calibrated, which is why having a scorer helps.
>
>
>
> ***[Do you believe that additionally optimizing for this can result in similar improvements to latency and an improvement in cost (ie, either replacing cost budget with latency budget or combining the two)?]***: Yes. Replacing the cost budget cost by the latency budget should work. For latency-critical applications, one can set up a small latency budget and always call the API with the fastest response rate.

---

### Meta-Review · Area_Chair_PWeJ · 2023-12-05

**Metareview:**

The reviewers appreciate the usefulness of the analysis provided and the framework as a whole. They also appreciate the presentation and clarity of the paper.

They do raise some concerns regarding lack of details, analysis, and ablations of the proposed method,

My main concern is that this might be better suited as a technical report rather than a scientific contribution.

**Justification For Why Not Higher Score:**

I did not get a clear picture of the scientific contribution of this work.

**Justification For Why Not Lower Score:**

N/A

---

### Decision · Program_Chairs · 2024-01-16

Reject